# Caring for Patients with Psychosis: Mental Health Professionals’ Views on Informal Caregivers’ Needs

**DOI:** 10.3390/ijerph18062964

**Published:** 2021-03-14

**Authors:** Maria Moudatsou, Sofia Koukouli, Eleutheria Palioka, Garyfalia Pattakou, Panagiota Teleme, Georgia Fasoi, Evridiki Kaba, Areti Stavropoulou

**Affiliations:** 1Department of Social Work, School of Health Sciences, Hellenic Mediterranean University, 71410 Heraklion, Greece; koukouli@hmu.gr (S.K.); palioka94@hotmail.com (E.P.); garyfaliaptk@outlook.com.gr (G.P.); giwtateleme@yahoo.gr (P.T.); 2Laboratory of Interdisciplinary Approaches for the Enhancement of Quality of Life (Quality of Life Lab), Hellenic Mediterranean University, 71410 Heraklion, Greece; gfasoi@uniwa.gr (G.F.); ekaba@uniwa.gr (E.K.); astavropoulou@uniwa.gr (A.S.); 3Centre of Mental Health, 71201 Heraklion, Greece; 4Department of Nursing, School of Health and Care Sciences, University of West Attica, 12243 Athens, Greece

**Keywords:** informal caregivers, mental health professionals, psychotic syndrome, patients, family, health needs

## Abstract

The aim of this study was to explore the views of mental health professionals regarding the needs of the informal caregivers of patients with chronic psychotic syndrome. A qualitative research design was used. The sample consisted of 12 mental health professionals selected by a purposive sampling strategy. Data were collected through semistructured, face to face interviews. Framework analysis was used to analyze qualitative data and establish main themes and subthemes. Three main themes emerged namely, (i) impact of caring on caregivers’ lives, (ii) caregivers’ needs, and (iii) recommendations for better care. Informal caregivers’ needs were conceptualized into subthemes within the main themes. Caregivers’ increased responsibilities of caring for their relatives, the impact on their mental and physical health status and the restrictions in their social and professional life were revealed. Targeted health interventions and social policy planning are recommended for supporting informal caregivers and improving patient care.

## 1. Introduction

In recent years much emphasis was placed upon the psychiatric reform, at an international level, including the community based care of people with psychosis [1,2,3,4]. Within the context of this reform—which was initiated in Greece in 1984—there was a focus on the social integration of people with mental disorders and thus new roles were given to the family caregivers [5]. The deinstitutionalization of patients and the reduction of in-hospital care have resulted in increased caring responsibilities for family caregivers [6]. The value of community based care for patients with psychotic syndrome has been highlighted in relevant literature as it is considered fundamental in reducing the number of hospitalizations and enhancing patient’s psychosocial function [7,8].

A caregiver as a term refers to anyone who provides support to a person in need or a patient, both to meet his or her daily needs and for companionship [8]. Caregivers are divided into formal (providing care as a paid service) and informal (family, relatives, friends or neighbors) who provide unpaid daily support to a person in need [9].

According to the social model of health, need is a complex concept, which includes a wide range of issues related to the wider social and economic environment in which one lives, such as personal health, housing, health services, education, work, financial situation, and others [10,11,12].

Informal caregivers are concerned with patients’ needs and abilities to take control of their environment and deal with stressful situations [11,12,13]. The multiple roles and the increased responsibilities of informal caregivers impact on their self-esteem, coping ability and problem-solving capacity [14,15]. This is even more evident when care is provided to patients with psychotic syndrome. According to the relevant literature, the caregivers of these patients often experience helplessness, loss of control, guilt, and low self-esteem [16,17].

Chronic stress experienced by caregivers who have taken on the responsibility of a dependent relative, affects their health in three distinct areas: the self-reported health status, the health related behaviors such as smoking, use of medicine, and their ability of making use of health services to meet their own health needs [18]. In addition to that, the stress experienced by caregivers has been found to be associated with poor immune function, low levels of physical and mental health, as well as high mortality rates [19]. Moreover, being the main informal caregiver of a patient with psychotic syndrome leads to poor social interaction [20].

The needs of the caregivers include objective needs related to the patient’s behavior and needs associated with the social, economic and family burden as well as with health and leisure issues [14]. Caring for a person with chronic psychotic disease is so demanding in time and energy that their caregivers are forced to reduce working hours, or give up their job. This is a significant issue as work and professional engagement appear to have an empowering role on caregivers’ life and wellbeing [21].

The caregiver’s engagement with social activities or leisure activities can help them to cope with heavy caring tasks [13]. However, as the review of literature demonstrates, caregivers of patients with psychotic syndrome experience restrictions in daily activities, social life and leisure [22]. In addition, social prejudice and stigmatization impact on their personal and professional life, creating feelings of embarrassment and guilt [23].

Educational interventions on coping strategies, stress management and family’s relationships and interactions are referred to as necessary actions for caregivers’ empowerment [24]. In addition, interventions for both patients and families include assessment of patients and caregivers’ health needs, psycho education, handling emotions and interpersonal relationships as well as patients’ rehabilitation programs [25,26,27].

Finally, the economic restraints and the resulting limitations in community mental health services may cause major consequences and burdens to the family caregivers of the mentally ill patients [2,28,29].

The needs of the caregivers of patients with chronic psychotic syndrome is a research area that is not thoroughly investigated within the Greek context [5,28,29]. This raised the necessity to shed light in this particular subject. The healthcare professionals’ opinions about the informal carers’ needs and their interpretation is of particular importance especially in the field of mental illness, which is not easily accessible by the researchers. In the present study it was imperative to collect this information from the healthcare professionals as the caregivers themselves were rarely available. Furthermore, exploring the health professionals’ views regarding the needs of the mental patients’ caregivers is even more important as it brings to the fore aspects as experienced by the formal carers and remain to a certain extent unexplored.

## 2. Material and Methods

### 2.1. Aim

The aim of this study was to explore the views of mental health professionals regarding the needs of the informal caregivers of patients with chronic psychotic syndrome.

### 2.2. Methods

#### 2.2.1. Study Design

A qualitative case-study research design based on framework analysis [30], was applied in the present study. Qualitative research approaches allow the in-depth examination of people’s views, experiences, perceptions, and feelings. They further allow the researcher to identify issues from the participants’ perspective and understand the meanings and interpretations that they give to certain phenomena or events [30]. This methodological approach may provide in-depth knowledge on how healthcare professionals assess the needs of caregivers of patients with chronic psychotic syndrome, a topic which has not been thoroughly examined in Greece.

#### 2.2.2. Settings and Sample

A purposeful sampling strategy was used. The use of a purposive sampling technique leads to a sample of people that fulfill certain criteria chosen by the researcher. It is referred as a strategic way to sample individuals who are relevant to the research question posed and enables the researcher to identify potential participants who may provide a wealth of information [31].

Mental health professionals who work in public healthcare services in an urban city of northern Greece consisted the study population. In specific, social workers, psychologists, and psychiatrists, who were employed in three tertiary care hospitals and four primary healthcare centers, having at least one year of working experience with caregivers of patients with chronic psychotic syndrome were included in the present study. This working experience was considered necessary in order for the potential participants to be able to provide sufficient information on the topic under investigation. It is important to notice, that it is an ordinary practice in Greece that people with mental illness visit healthcare professionals (e.g., psychiatrists, psychologists, social workers) accompanied by their informal carers, in order to support them during their therapy. Therefore, caregivers inform health professionals not only for their patients’ condition but also for their caring burden and their interaction with the patient. This is due to the lack of a well-organized community care system for patients with psychosis in Greece. Therefore, caregivers undertake the main responsibility of caring for the patient at home.

Recruitment of the potential participants took place within the healthcare services by the researchers. Initially, the Directors and the Health Managers of each site were informed about the study. Information was then distributed to mental health professionals who were asked to contact the researchers if they were interested in participating in the study. Twelve mental health professionals agreed to participate in the study.

#### 2.2.3. Data Collection

Semi-structured, face to face interviews were used for data collection. This method of interviewing is largely used in qualitative research, allowing the researcher to thoroughly explore the participants’ thoughts and beliefs [32]. A flexible interview scheme was used involving four open-ended questions addressing the following areas: (a) the feelings and responsibilities of the caregivers, (b) the needs of the caregivers, (c) the degree to which these needs are met, and (d) the ways that these needs might be effectively fulfilled in the future. The above areas of exploration were formulated following an attentive review of the relevant literature as well as experts’ opinion. Follow-up questions, probes and comments were encouraged throughout the interview course. This method allowed the researcher to collect open-ended data and to explore the participants’ thoughts and beliefs regarding the needs of caregivers of patients with chronic psychotic syndrome. Reflexive notes were also kept throughout the course of interviews. These notes helped the researchers to gain in-depth understanding of the participants’ perceptions and used to foster trustworthiness of the research process, as described in Section 2.2.4.

The research team consisted of university teachers and experts from different scientific areas (e.g., nursing, social work, sociology) with experience in the care of people with mental illness and in qualitative research. Specifically, the team consists of five professors, with vast experience in designing and carrying out qualitative research. Most of them have long-term experience in working with vulnerable groups of people or with people with mental illness in the community and in psychiatric settings.

Three members of the research team, who were well-trained in conducting interviews, interviewed the study participants. Although they had experience as social workers in mental health care, they did not work in any of the involved study sites and had no established relationship with the participants prior to the study. This was considered an advantage for the study as the participants could speak freely about their views on caregivers’ needs. The location and time of interviewing were chosen by the respondents. Protection of the study participants, reassurance of privacy and diminishing the possibility of the discussion to be heard or seen by other people were prerequisites for selecting the location of the interview [33]. Thus, a secluded room inside the healthcare service was chosen. For data collection purposes, a tape recording was used. Interviews lasted 50–80 min. The collected data were transcribed verbatim by the interviewers, right after the completion of each interview.

#### 2.2.4. Data Analysis

The method used to analyze the data obtained from the interviews was the framework analysis. This is a method that consists of five distinct, interconnected, steps: (a) enable the researcher to understand and interpret the data, (b) extract codes that describe the information contained in the data, (c) extract broader categories and specific themes, (d) structure the data based on the issues raised, and (e) writing the results in a coherent way that gives meaning to the data [34]. Framework analysis is an appropriate method widely used in the analysis of qualitative research data and can be conducted by multidisciplinary research teams including healthcare professionals, psychologists and sociologists. It forms a systematic and rigorous method of analyzing data that enables teams of researchers to work together. The interdisciplinary collaboration required in this method of analysis calls for a critical, extensive and reflexive dialogue between the researchers that come from different disciplines, enhancing thus the credibility and relevance of the findings [35]. Based on these, framework analysis was considered the best possible choice for analyzing the data of the present study.

Finally, trustworthiness of the study was ensured by applying specific strategies, such as analyst triangulation, peer review and reflexivity. Data were analyzed independently by two members of the research team. Interpretations were compared and discussed until consensus was reached on the most representative ones. In addition, the study processes were evaluated by peer investigators—academics who were experienced in qualitative research and reflexive notes were kept by the researchers regarding the settings and the interview process [36].

#### 2.2.5. Ethical Consideration

Prior to the commencement of the study, ethical approval was gained by the Ethical Committee of the Hellenic Mediterranean University as well as the involved organizations (Ref. no 1029/25 May 2017). Application of appropriate ethical principles and protection of human subjects were preserved throughout the course of the study [33]. Before the commencement of the interview, the participants were fully informed about the aim and the nature of the study. Permission was also granted for the use of tape recorder. Informed consent forms were signed, and issues of anonymity, confidentiality and voluntary participation were confirmed by the researchers. The participants were also assured that the data will be used for scientific purposes only and the study findings will be presented in a way that will not disclose any ID information.

## 3. Results

Twelve health professionals working in public mental healthcare services (primary and tertiary) in an urban city of northern Greece participated in the study. Eight out of them were social workers, while the other four were psychologists (2) and psychiatrists (2). Their working experience ranged from 4 to 34 years, while their experience with caregivers of patients with chronic psychotic syndrome ranged from 1 to 30 years. Sample characteristics are presented in Table 1.

Framework analysis led to the formulation of three main themes, namely, (i) impact of caring on caregivers’ lives, (ii) caregivers’ needs, and (iii) recommendations for better care. Informal caregivers’ needs were conceptualized into subthemes within the main themes i, ii, and iii. Main themes and subthemes are presented in Table 2.

### 3.1. Impact of Caring on Caregivers’ Lives

Mental health care professionals viewed that caring for a person with psychosis differs significantly from caring for a physically ill but mentally healthy person. They referred to the unique nature of mental care and highlighted that caring for a relative with chronic psychotic syndrome creates increased responsibilities for the caregivers. They particularly highlighted that some of the caregivers’ responsibilities included follow up and urgent care visits to the doctor and provision of multifaceted care, such as medication taking, personal hygiene, maintenance of mobility, and social skills development. These caring issues appeared to have a significant impact to caregivers’ lives, as most of them undertake the full responsibility of the patient’s care. Data analysis revealed that caregivers of patients with chronic psychotic syndrome seem to experience restrictions in various areas of their lives, namely, financial and professional burdens, social constraints, physical and mental deterioration.

#### 3.1.1. Financial and Professional Burdens

The participants referred to the financial and professional burdens that the caregivers of patients with chronic psychotic syndrome face. The degree to which their financial and professional life is affected varied according to the degree of dependency and the severity of their relative’s illness. Quite a few of the study participants argued that caregivers find it difficult to balance their professional obligations with those resulting from patient’s care. This results in a significant reduction of their working hours, or, in case that the caregivers are the parents of a child or an adolescent with mental illness, then one of the parents has to leave his/her job to take care of the child. In this case, the whole family is affected, as its socioeconomic status is modified.

*“The caregivers who work are those who have functional patients* (with psychotic syndrome)*”* (p. 1).

*“Usually what happens with caregivers who are parents, one of them has to leave the work and stay at home. This changes the social and financial profile of the family”* (p. 3).

Other participants, however, argued that caregivers seemed to be able to find the right balance between their professional obligations and the patient’s care.

*“I think this is usually settled…There are day care centers, where patients can go while their relatives are at work”* (p. 4).

*“We rarely see someone to leave his job in order to take care of the patient”* (p. 10).

Changes in professional status consequently lead to financial restrictions as well. In addition, caregivers often have to either support financially the patients, as the latter do not work, or cover part of their expenses, such as hospitalization. Finally, the participants emphasized that, even when caregivers are able to maintain their jobs, family budgets are often reduced due to the costs related to patient’s care.

*“Caregivers face financial difficulties, because patients with chronic psychotic syndrome often can’t work…their medication is expensive…caregivers are financially hampered…”* (p. 5).

*“They* (the caregivers) *experience financial problems as they have to cover their relatives’ expenses, and they* (the relatives) *have a lot… they smoke too much, they have no insurance and their medicines are so expensive…”* (p. 11).

#### 3.1.2. Social Constraints

Data revealed that several factors related to patients’ care and increased responsibilities impact caregivers’ social lives. The participants focused on the unique nature of the illness that is related to social restrictions and problems. For example, issues of psychological exhaustion, overworking, negative feelings, abrupt mood changes, and stigma, may lead to social isolation. The social lives of caregivers seem to be remarkably restricted due to the above factors, even for people who are well educated.

*“….there is too many restrictions* (in caregivers’ social life)… *I think their social relationships are few, because they don’t have enough time… they are not in a good mood”* (p. 1).

*“They are socially isolated, a major reason is that they feel shamed … even educated people do not accept the situation”* (p. 6).

In addition, many of the participants referred to social stigma, as an issue that negatively affects social relationships and leads to self-isolation. They considered that social isolation is more due to stigma than to increased caring responsibilities. They further stressed that, despite the sensitization campaigns and other social interventions made to diminish social stigma, mental illness still breeds fear and rejection.

*“…mental illness remains an issue that raises many fears… causing defensive and rejection behaviors”* (p. 3).

*“I think that they* (the caregivers) *are socially constrained not so much because of their responsibilities, but because of the stigma they are experiencing. Many caregivers prefer to keep it (the illness) secret…”* (p. 10).

#### 3.1.3. Physical and Mental Deterioration

Participants referred to the physical and mental health deterioration that the caregivers experience. Uncertainty regarding appropriate decision making for caring issues, stress, anxiety, and self-despair may result in health decline. In addition, caring responsibilities, physical and mental exhaustion and an array of negative feelings that the caregivers experience, seem to impact negatively on their physical and mental health.

*“They* (the caregivers) *have a lot of psychosomatic problems… headaches, migraines, skin eczema, it is a physical and mental exhaustion…”* (p. 1).

*“They are stressed, they develop problems such as diabetes, hypertension, and angina…it is the anxiety, the responsibilities, the guilt …”* (p. 3).

*“Physical health can also be hampered because they* (the caregivers) *neglect themselves… the regular checkups they have to do… the patient is always a priority”* (p. 9).

In addition, the participants referred to feelings of depression and low self-esteem as having a serious impact on the caregivers’ mental health status.

*“It’s changing,* (the caregiver’s) *mental status is changing a lot. Usually, they become anxious and depressed”* (p. 3).

*“I would say that they have low self-esteem… it depends on personality traits, educational level, financial status…. but in general, their self-esteem is negatively affected”* (p. 1).

Furthermore, the participants referred to the negative feelings that the caregivers experience. Chronic stress, anger, grief, hopelessness were considered as extra burdens that affect the caregivers emotionally.

*“Many times they feel guilty of whether they handled the situation properly or not…”* (p. 4).

*“They feel angry …it may not be seen at a first glance…they manage to hide it, but the patients feel it”* (p. 6).

*“They feel sad, despaired, depressed”* (p. 5).

### 3.2. Caregivers’ Needs

The second main theme that emerged from the data analysis concerned the needs of the caregivers. Increased caring responsibilities, financial and professional burdens and health decline as previously reported, generated an array of needs for the caregivers. The mental health professionals who participated in the study referred to the needs which remain unmet by the healthcare services. These needs were framed by issues related to economic and psychosocial support, as well as information giving and consultation.

#### 3.2.1. Economic Support

The demanding caring roles of the caregivers may cause severe financial consequences for the whole family. As previously reported many caregivers of patients with chronic psychotic syndrome often face financial strains, either due to their inability to work or due to the increased healthcare expenses. Reductions of their monthly income, increased out of pocket expenses, work related costs due to labor dropout, and insufficient allowances are strains that impact to caregivers’ life and wellbeing.

*“Financial support is needed, because the allowance that patients receive is too small to live with”* (p. 9).

*“Caregivers need support…mainly financial…to feel a bit better for a while”* (p. 10).

In addition, participants pointed out the substantial role of the caregivers and the necessity for supporting them financially.

*“Financial support is important for the caregivers…they are doing something very substantial… keeping a patient* (with psychotic syndrome) *in the community…a very important role for the caregiver that should be supported”* (p. 1).

#### 3.2.2. Psychosocial Support

Caring for a person with a chronic psychotic syndrome raises multiple problems for informal caregivers. Coping and adapting at a psychosocial level is not always easy especially when issues of social stigma and ineffective self-control are involved due to patients’ condition. Many participants referred to the need for psychosocial support, as seen to be a priority for the caregivers. A collaborative approach between caregiver and health professionals, group psychotherapy and self-help groups involving families that face similar situations, may address these problems. In addition, the development of self-control skills was mentioned by the participants as they may limit impulsive behavior and negative feelings.

*“They* (the caregivers) *need group psychotherapy… To be together with families which face the same problem…to receive advice and support”* (p. 11).

*“… they* (the caregivers) *acted as a self-help group…it is somehow a self-healing group”* (p. 3).

*“Self-control…I believe that having* (the caregivers) *self-control can help a lot”* (p. 12).

#### 3.2.3. Information Needs

Supporting the caregivers in terms of consultation, referrals and information on the disease management was considered essential for alleviating feelings of helplessness and uncertainty. The participants highlighted the caregivers’ needs for getting information and guidance mostly on practical matters of care and appropriate use of healthcare services. By addressing these needs a beneficial effect may occur for both the patient and the caregiver.

*“To get practical advice… and to feel that they are not alone in it”* (p. 1).

*“They* (the caregivers) *have mainly psychological and informative needs…to know where they can refer to, what they can ask for, what they are entitled to…”* (p. 3).

### 3.3. Recommendations for Better Care

Recommendations for better care was the third main theme that emerged from the data analysis in the present study. This theme involved the health professionals’ views on how to address the unmet needs of the caregivers and how to improve the healthcare services provided to persons with mental illness. Appropriate interventions for meeting the caregivers’ needs and for improving the quality of healthcare services may have a direct positive impact on patients and caregivers. They may also support mental healthcare professionals themselves to provide better care, focused not only on the patients but on the caregivers too.

#### 3.3.1. Meeting the Caregivers’ Needs

A significant issue raised by the study participants concerned the caring priorities set by the mental health care professionals. These priorities focus on the treatment of illness and on fulfilling the patients’ needs rather than addressing the needs of the caregivers. To this end, caregivers remain alone and unsatisfied as most of their needs are neglected. The health professionals seem to recognize this flaw and efforts are made for covering caregivers’ needs as much as possible.

*“…we all* (the mental health professionals) *tend to pay attention on the patient, we rarely focus on the caregivers’ burden”* (p. 4).

*“…we* (the healthcare service) *have group therapy for the patients, but not for the caregivers…“* (p. 2).

Mental health professionals referred to their efforts to get in contact with self-help organizations as they advocate that such organizations have a significant impact on fulfilling the caregivers’ needs.

*“What I know is that this self-help organization does a very good job and we are in contact with them”* (p. 9).

#### 3.3.2. Improvement of Services

Quality improvements and restructuring of the healthcare services were mentioned by the study participants as critical issues for providing better care in the context of mental health. Organizational innovations, novelty in policy making, involvement of stakeholders in organizational planning were recommended actions for better care. Most of the participants stressed that it is important for the caregivers to be involved in healthcare policy making. A reorganization of the services that will address the caregivers’ needs was also underscored.

*“Of course they* (the caregivers) *should be involved in the restructuring of the services provided…because they are the ones who know what they really need”* (p. 2).

*“…it is self-evident that the interested stakeholders must be involved in policy making”* (p. 3).

In addition, the participants argued that adequate funding and staffing of the healthcare services are important factors to improve the care provided to patients and caregivers.

*“If there was adequate funding and appropriate staffing, it could be possible* (to improve care)*”* (p. 11).

Finally, many of the participants emphasized the need to improve the existing services by introducing quality initiatives and specialized, tailor-made services to caregivers.

*“A more organized context is needed… specialized services focused on caregivers, something new and innovative…”* (p. 1).

*“Improved services, quality oriented, and focused on the needs of caregivers”* (p. 3).

## 4. Discussion

The results of our study suggested that caring for patients with chronic psychotic syndrome has a direct impact on their informal caregivers’ daily life. Personal and family life are affected to great extent as financial and professional constraints, social deterioration and health decline are emerging problems within the context of caring for a mentally ill relative. Caring for a patient with psychotic syndrome was referred to as a family burden in many research studies [37,38]. Psychosis affects caregivers’ lives not only in the early stages of the disease but throughout its development and the occurrence of acute episodes, calling for increased caring responsibilities and demanding caring roles [39,40]. Parents and especially mothers of psychotic patients appear to be overwhelmed by their caring role and responsibilities [41]. As a result of this distressing role, caregivers of psychotic patients experience intense feelings of loss, sadness, grief, and emotional exhaustion [42]. Research evidence emphasized that caregivers of people with psychosis experienced more intensively feelings of pain and loneliness than caregivers of other patient groups [42,43,44], which is in agreement with the findings of the present study. Furthermore, feelings of stress, anxiety and self-despair—which impact negatively on caregivers’ physical and mental health—were referred to as predominant feelings for the caregivers as most of them seemed uncertain of whether or not offer the proper care and treatment to their relatives. This is in accordance with a study which stated that informal caregivers often feel guilty and blame themselves for not providing the appropriate care and support to their relatives [45].

In many cases, the caregiver is required to act as a mediator among the patient and the healthcare services or to handle bureaucratic and organizational issues which are related to patient’s care. Additionally, the caregivers supervise the patient in order to ensure that the medication is taken correctly and personal and home hygiene is maintained. Caregiving for a mental health patient appears to be a highly demanding responsibility involving vigor, knowledge, empathy, and financial sustenance [46]. In the same line to the results of the present study, another study stresses that caregivers’ duties were demanding as they are targeting on both, meeting patients’ health needs and preserving their well-being [47]. Mental disorders’ symptoms and patient’s behavior might restrict the caregivers’ functional capacity, increase their responsibilities in activities of daily living and disrupt their household routine [48,49].

This study highlights that the demanding caring roles lead to the deterioration of professional, financial and social life of the caregivers. Additionally, caregivers’ physical and mental health are adversely affected. More specifically, caring for mentally ill patients impacts negatively on the caregivers’ professional and financial status as many of them appear to leave their job or reduce their working hours. Relevant research evidence advocates that informal caregivers of patients with chronic psychotic syndrome face professional and financial problems that impact severely on their daily living [50].

Moreover, caregiving for mentally ill patients appears to have negative consequences on peoples’ social lives. Caregivers are often isolated from their social environment due to time constraints or feelings of shame and guilt. Social stigma was also referred to as a reason of social isolation. These results are consistent with those derived from previous research, which suggested that caregivers of patients with chronic psychotic syndrome tend to withdraw from their social environment [22,36,51,52,53]. The results of these studies attributed caregiver’s social isolation to either lack of free time or social stigmatization [22,23,49,54]. Additionally, in contrast with the results of the present study, findings from previous research stressed that social isolation may result in family dysfunction [36,51].

According to health professionals the caregivers of patients with chronic psychotic syndrome develop many mental and physical health problems. Study participants emphasized that these problems are related to stress, physical and mental exhaustion, depression, self-despair, anger, and grief that carers experienced due to their caring responsibilities for the patient. Furthermore, lack of leisure time and neglecting their own health appear to have a significant negative impact on caregivers’ health. Relevant studies advocate that caring for mentally ill patients and related anxiety impact negatively on caregivers’ health and result in a physical and mental health decline [23,52,55,56,57,58,59,60,61,62,63,64]. Especially, caregiving for psychiatric patients increase mental health problems, levels of anxiety and depression for the informal carers, as well as the risk of burnout syndrome [36,51].

The results of the present study described three main areas on caregivers’ needs, namely, economic, psychological support and information giving. Economic support was considered as a crucial issue for the informal caregivers, as they face many financial problems that impact negatively on their daily life. This was supported by relevant studies which underscored the need of specific policy interventions and financial support for mentally ill patients and their families [20,50,65,66]. Furthermore, a well-organized supportive framework regarding the psychological support of caregivers was highlighted by the participants. This may enable them to express their fears and feelings and support caregiving tasks more efficiently [50,67,68].

Information giving is considered a necessity for effective caring and as such special emphasis should be placed on diffusion of information and informal caregivers’ education [24,50,69,70,71]. Recommendations for better care involved meeting the caregivers’ needs and improved health services. Caregivers’ dissatisfaction and inadequate provision of quality healthcare services were reported as areas that require attention and improvement. This is supported by relevant studies that underscore the health services’ insufficient support and the unmet caregivers’ needs that lead to dissatisfaction [2,50].

Improvements in existing healthcare services and assessment of caregivers’ needs at a primary healthcare level are recommended by relevant literature in terms of providing quality healthcare for this specific group of patients and their caregivers [2,4,50,54,72]. Additional social support programs, such as social skills training, group therapies, employment opportunities, access to day clinics, are necessary for both patients and families [68].

### 4.1. Limitations of the Study

Generalizability of the study results is not the expected outcome of the present study, as qualitative research designs do not aim at generalizing the results but at exploring the uniqueness of the phenomenon under investigation. One of the study’s limitations is related to the difficulty to recruit healthcare professionals working in sensitive contexts such as the psychiatric hospital wards. In this study the sample was scarce, deriving from the public healthcare sector of a single geographical area. It should be thus, expanded to provide more explicit and sound information. Results should be viewed under this limitation.

### 4.2. Policy Implications and Recommendation for Future Research

The results of the present study may contribute to the enrichment of the existing knowledge in the field of care for patients with chronic psychotic syndrome. More precisely the study provides evidence on informal caregivers’ needs, through an in-depth exploration of the healthcare professionals’ views. Mental health care professionals are the key-stakeholders, having a distinct experience regarding the informal caregivers’ needs due to their daily contact and interaction with them. In this respect, health professionals can provide unique knowledge on the carers’ needs and shape future direction for improved healthcare services [73,74]. Future research involving the informal caregivers themselves may provide further evidence for policy making and quality interventions in healthcare services. In addition, further research in this area is recommended involving health professionals employed in both public and private healthcare sectors and from broader geographical domains including urban and rural areas in Greece.

## 5. Conclusions

Τhe findings from the present study highlighted the necessity of supporting informal caregivers’ needs by implementing appropriate health and social policy strategies. Caregivers’ increased responsibilities, social and professional restrictions and impairments in their physical and mental health status have a severe impact on their daily life and patient’s care. Health and social care interventions and educational programs, at individual, family and community levels, are recommended for supporting informal caregivers. Social policy planning for effective interventions in prevention, treatment and rehabilitation of patients with chronic psychotic syndrome should include the views of both the patients and the caregivers. Further research is recommended in exploring the needs of the caregivers of patients with chronic psychotic syndrome, as perceived by the caregivers themselves, in order to have a more integrated view of the subject under investigation.

## Figures and Tables

**Table 1 ijerph-18-02964-t001:** Characteristics of the study sample.

a/a	Specialty	Years of Employment in Mental Health Services	Years of Working Experience with Informal Caregivers’ of Patients with Psychotic Syndrome	Type of Employment Service
1	Social Worker	15	1	Community healthcare center A
2	Social Worker	21	21	Community healthcare center A
3	Social Worker	24	24	Community healthcare center B
4	Psychiatrist	7	7	Hospital A
5	Social Worker	27	27	Hospital B
6	Social Worker	33	30	Hospital B
7	Social Worker	13	13	Community healthcare center C
8	Psychiatrist	8	8	Community healthcare center D
9	Social Worker	10	10	Community healthcare center B
10	Social Worker	34	4	Hospital A
11	Psychologist	3	3	Hospital C
12	Psychologist	22	22	Hospital C

**Table 2 ijerph-18-02964-t002:** Main themes and subthemes.

Main Themes	Subthemes
1. Impact of caring on caregivers’ lives	1.1. Financial and professional burdens 1.2. Social constraints 1.3. Physical and mental deterioration
2. Caregivers’ needs	2.1. Economic support 2.2. Psychosocial support 2.3. Information needs
3. Recommendations for better care	3.1. Meeting the caregivers’ needs 3.2 Improvement of services

## Data Availability

Data generated during the present study is not possible to be shared due to issues of subjects’ privacy and confidentiality.

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
