# Peer review of "Caring for Patients with Psychosis: Mental Health Professionals’ Views on Informal Caregivers’ Needs"

_ijerph, 2021, doi:10.3390/ijerph18062964_

Round 1

Reviewer 1 Report

Thank you for a well written, topical piece of work. Overall, I found the study to be clearly presented and interesting. Some minor points to consider:

  1. Why is it important that you collect this information from health professionals rather than from the care-givers themselves? This seems to be a limitation and is not adequately described, particularly in lines 83-86. Some further reference to how health professionals views and opinions are most appropriate would be beneficial.
  2. It would be useful to clearly describe what "work experience with care-givers" actually means. Do they counsel them? Do they work with them informally in different capacities? This would help your justification of why you chose to interview health professionals rather than care-givers in the first instance. E.g.: are you only exploring the views? Or are you looking at experiences as well? 
  3. Lines 308 - 310 should be italicised
  4. I thought your results were well presented and the quotes supported your findings.
  5. Your discussion and conclusions were also well described and the higher level thinking was evident, particularly hen discussing the potential for future interventions. 

  1.  

Author Response

REPLIES TO THE REVIEWER’S COMMENTS

The authoring team would like to thank the reviewer for his/her constructive remarks. Revisions were made according to the reviewers’ comments.

Replies to reviewer’s comments are following:

REVIEWER 1

Thank you for a well written, topical piece of work. Overall, I found the study to be clearly presented and interesting. Some minor points to consider:

  1. Why is it important that you collect this information from health professionals rather than from the care-givers themselves? This seems to be a limitation and is not adequately described, particularly in lines 83-86. Some further reference to how health professionals views and opinions are most appropriate would be beneficial.

REPLY

In the introduction we have added that: 

The health care professionals’ opinions and interpretation on informal carers’ needs is of particular importance especially in the field of mental illness which is difficult for the researchers to approach.

We have also added in the Policy Implications section that:

Mental health care professionals are the key-stakeholders, having a distinct experience regarding the informal caregivers’ needs due to their daily contact and interaction with them. In this respect, health professionals can provide a unique knowledge on the carers’ needs and shape future direction for improved health care services. In the present study it was imperative to collect this information from the health care professionals as the caregivers themselves were rarely available.

Relevant reference have been included at this point as per suggestion.

We would like also to say that we used a sample of health professionals as a valuable alternative view / perspective regarding the needs of informal caregivers and not because we considered their views as most appropriate. 

  1. It would be useful to clearly describe what "work experience with care-givers" actually means. Do they counsel them? Do they work with them informally in different capacities? This would help your justification of why you chose to interview health professionals rather than care-givers in the first instance. E.g.: are you only exploring the views? Or are you looking at experiences as well? 

REPLY

We have added in Setting and Sampling section: 

It is important to notice, that it is a common practice in Greece, people with mental illness to visit health care professionals (e.g. psychiatrists, psychologists, social workers) accompanied by their informal carers, in order to support them with their therapy. So, caregivers inform health professionals not only for their patients’ condition but for their caring burden and their interaction with the patient. This is due to the lack of a well-organized community care system for patients with psychosis in Greece. Therefore, caregivers undertake the main responsibility of caring for the patient at home.

  1. Lines 308 - 310 should be italicised

REPLY

We have corrected that.

  1. I thought your results were well presented and the quotes supported your findings.

  1. Your discussion and conclusions were also well described and the higher level thinking was evident, particularly hen discussing the potential for future interventions. 

Thank you very much!

The Authoring Team.

Reviewer 2 Report

Thank you very much for the opportunity to review this article. Next, I will break down this review:

First of all, I consider that the sample of participants is small for a qualitative study. The sample size in qualitative research is determined by redundancy of information or saturation of the interview data, which is the criterion that the authors should have followed. The saturation point in this type of study is usually achieved between interviews 20 and 25 (Turner-Bowker et al., 2018), with authors who recommend conducting two or three more interviews to confirm that saturation has been reached. (Moser and Korstjens, 2018). Regarding the limitations of the study, the authors indicate the difficulty of recruiting professionals from the psychiatric field where they carry out the study, but even so, I consider that the sample is somewhat insufficient, although perhaps it could arouse the interest of other authors to expand knowledge about the phenomenon study.

Turner-Bowker, D. M., Lamoureux, R. E., Stokes, J., Litcher-Kelly, L., Galipeau, N., Yaworsky, A.,… Shields, A. L. (2018). Informing a priori Sample Size Estimation in Qualitative Concept Elicitation Interview Studies for Clinical Outcome Assessment Instrument Development. Value in Health, 21 (7), 839–842. doi: 10.1016 / j.jval.2017.11.014.

Moser, A., and Korstjens, I. (2018). Series: Practical guidance to qualitative research. Part 3: Sampling, data collection and analysis. European Journal of General Practice, 24 (1), 9-18. doi: 10.1080 / 13814788.2017.1375091.

On line 60, it is recommended to replace “external determinants of health” with “social determinats of health”.

On line 94, the authors must indicate what type of qualitative study design they have carried out (phenomenology, qualitative case study, etc.). In addition, when they indicate the analysis framework on which they have been based “framework analyzes” they must place the reference number below. On the other hand, during the text, sometimes they write “framework analysis” with a capital letter and others with a lowercase letter, they must unify the way of writing it.

On line 122, the authors indicate that semi-structured interviews were conducted using 4 open in-depth questions on 4 predetermined areas. They should explain how they decided on the previous areas of exploration, if they did so by means of a bibliographic review, consensus among researchers, prior surveys of professionals or a different method. In addition, they should indicate whether the researcher's field notes were made during the interviews and whether these notes were analytical material or were discarded and why. Although on line 168 they indicate that reflective notes were taken, they should indicate if they are field notes and provide more information on how they contributed to the research process.
Within the different quality criteria governing qualitative research studies (COREQ and SRQR), one of the relevant aspects that ensure the transparency of the study is the description of the research team. Authors must include a section that describes the credentials of the research team at the time of the study (profession, experience in qualitative research, experience in psychiatry or mental health, reflexivity, previous position regarding the phenomenon of study, etc.). They must also specify which team members conducted the interviews and who analyzed them, and who are the peer investigators who reviewed the investigation process (line 168). In addition, it is recommended to review both guides on quality criteria of qualitative studies to improve the methodological section of the article:

Consolidated criteria for reporting qualitative research (COREQ): a 32-item checklist for interviews and focus groups | The EQUATOR Network (equator-network.org)

Standards for reporting qualitative research: a synthesis of recommendations | The EQUATOR Network (equator-network.org)

Between lines 140-148, the authors describe ethical criteria that should be placed in their own section called “ethical considerations” or similar, but not within the “data collection” section.

On line 147, they must include that the study has been approved by the ethics committee indicated on line 484 and not only by the participating entities.

In line 177, Table 1, should provide more data on the participants, in order to favor the transferability of the study. For example, the type of service each participant works on.

In line 183, table 2, the two main topics called “caregivers´ feelings” and “caregivers´responsabilities” must go with their corresponding categories, as the following three topics appear. Therefore, the results 3.1. and 3.2. they should be rewritten according to the categories or subtopics obtained in the analysis. If there were no categories or sub-topics, it would indicate that the analysis has not been carried out correctly and / or that the information obtained is not enough to consider them independent topics.

From line 184 onwards, the explanation of the results obtained in each topic is scarce, the authors must describe each of the topics in more detail before providing narrative examples. In addition, the narrative examples must match the description. For example in line 188 they use the word "guilt" in the description and do not provide any narrative example of this feeling. They should review it on all topics. Before including narratives, it is not necessary for them to constantly repeat phrases such as "These were high through the following quotes." It is assumed that being a qualitative study, after the description of a result there will be narrative examples and including this phrase in all topics is repetitive. In addition, only the narrative of the participants should be in italics and between quotation marks, not the participant number or the explanatory notes that appear interspersed in some narratives (for example, in line 205 (the caregivers)).

Between lines 237-239, the authors name the categories identified from topic 3.3. differently than they do in table 2. The name of the categories must be the same. In addition, the categories included should be numbered to facilitate identification in the text. For example: 3.3.1. Financial and professional burdens. 3.3.2. Social constraits. And from then on follow this numbering. Each category, as well as each topic of all the results must be described more exhaustively before providing narrative examples.
The description of the category “Health deterioration” overlaps with the description of topic 3.1. “Caregivers' feelindgs”, providing similar results. Authors must differentiate results and not provide repeated results.

Between lines 308-310, narratives must be in italics.

Between lines 337-342, narratives should not go with classifier points.
In the discussion section, references are missing. They should include the reference after each sentence and not just after several sentences in a row. For example, at the beginning of line 376, they must include the reference after this sentence: “More specifically, caregivers of psychotic patients experience intense feelings of loss, sadness, grief, and emotional exhaustion for patients with psychosis”. They should review the entire discussion to correct these errors, for example they do it again between lines 384-387.
The references in the discussion are not referenced correctly. For example, on line 378 you should replace [42, 43, 44] with [42-44]. In line 408 they must substitute [22, 51, 36,52, 53] for [22, 36, 51-53]. They should correct these kinds of mistakes throughout the discussion.

The claims the authors make between lines 395-399 are incorrect. They cannot claim that there is an association between the caregiver role and the deterioration of the caregivers. Qualitative research does not establish associations between variables. They should reformulate indicating that the professionals describe this situation, but it is not an association in itself. The following sentence should be rephrased in a similar way and the authors should review the entire discussion and conclusions by rephrasing similar statements, such as lines 413-414: they cannot make that statement because it is not true that the results of their study indicate that caregivers develop health problems, but professionals describe this, which is different. The same, for example, in the conclusion of lines 463-465.

In the study limitations section, line 447, they should indicate the impossibility of generalizing the study results due to the methodological design used. In addition, what is described between lines 454-459 does not correspond to the limitations of the study, but to what its study contributes and with future lines of research, so it should go in a specific section and not in study limitations.

Finally, authors should check the references thoroughly, as there are many errors.

Considering the manuscript in a global way, I consider that the sample is scarce and that it should be expanded to provide greater depth to the information provided. Perhaps the little description of the topics and the lack of categories or subtopics in some of them may be the result of this. If this were possible, I think it is an interesting topic that deserves to be explored in greater depth.

A cordial greeting.

Author Response

REPLIES TO THE REVIEWER’S COMMENTS

The authoring team would like to thank the reviewer for his/her constructive remarks. Major revisions were made according to the reviewers’ comments.

Replies to reviewer’s comments are following:

REVIEWER 2

Thank you very much for the opportunity to review this article. Next, I will break down this review:

First of all, I consider that the sample of participants is small for a qualitative study. The sample size in qualitative research is determined by redundancy of information or saturation of the interview data, which is the criterion that the authors should have followed. The saturation point in this type of study is usually achieved between interviews 20 and 25 (Turner-Bowker et al., 2018), with authors who recommend conducting two or three more interviews to confirm that saturation has been reached. (Moser and Korstjens, 2018). Regarding the limitations of the study, the authors indicate the difficulty of recruiting professionals from the psychiatric field where they carry out the study, but even so, I consider that the sample is somewhat insufficient, although perhaps it could arouse the interest of other authors to expand knowledge about the phenomenon study.

Turner-Bowker, D. M., Lamoureux, R. E., Stokes, J., Litcher-Kelly, L., Galipeau, N., Yaworsky, A.,… Shields, A. L. (2018). Informing a priori Sample Size Estimation in Qualitative Concept Elicitation Interview Studies for Clinical Outcome Assessment Instrument Development. Value in Health, 21 (7), 839–842. doi: 10.1016 / j.jval.2017.11.014.

Moser, A., and Korstjens, I. (2018). Series: Practical guidance to qualitative research. Part 3: Sampling, data collection and analysis. European Journal of General Practice, 24 (1), 9-18. doi: 10.1080 / 13814788.2017.1375091.

REPLY

Although we understand the reviewer’s concern about the size of the sample and we have already mentioned that as a study limitation, we consider that the information gathered by the 12 participants provided us a thorough understanding of the subject under investigation. To the best of our knowledge choosing a suitable sample size in qualitative research remains an area of conceptual debate. Qualitative research experts argue that there is no straightforward answer to the question of ‘how many’ and that sample size is contingent on a number of factors relating to epistemological, methodological and practical issues [Baker SE, Edwards R. How many qualitative interviews is enough?: expert voices and early career reflections on sampling and cases in qualitative research. National Centre for Research Methods Review Paper. 2012]. In general samples in qualitative research tend to be small, as the aim of this form of study is not to estimate the prevalence of a phenomenon but to provide an in-depth understanding of a topic, to develop explanations and to generate ideas or theories. [Sandelowski M. One is the liveliest number: the case orientation of qualitative research. Res Nurs Health. 1996;19(6):525–9] .According to Mantzoukas in qualitative research the sample usually is a double-digit or even a single-digit number. A large sample can neither serve the aims nor the objectives of qualitative research, but instead works negatively for its validity, as in the case of a large number of participants the subjective and individualized characteristics are lost [Mantzoukas, S. Qualitative research in six easy steps. The epistemology, the methods and the presentation. Nursing 2007, 46, 88-98.].

On line 60, it is recommended to replace “external determinants of health” with “social determinats of health”.

REPLY

We have replaced that as per suggestion.

On line 94, the authors must indicate what type of qualitative study design they have carried out (phenomenology, qualitative case study, etc.). In addition, when they indicate the analysis framework on which they have been based “framework analyzes” they must place the reference number below. On the other hand, during the text, sometimes they write “framework analysis” with a capital letter and others with a lowercase letter, they must unify the way of writing it.

REPLY

We have now specified that we used a qualitative case-study research design.

We have also included the relevant reference right after the  framework analysis and we unified the way of writing “framework analysis” throughout the text as per suggestion.  

On line 122, the authors indicate that semi-structured interviews were conducted using 4 open in-depth questions on 4 predetermined areas. They should explain how they decided on the previous areas of exploration, if they did so by means of a bibliographic review, consensus among researchers, prior surveys of professionals or a different method.

In addition, they should indicate whether the researcher's field notes were made during the interviews and whether these notes were analytical material or were discarded and why. Although on line 168 they indicate that reflective notes were taken, they should indicate if they are field notes and provide more information on how they contributed to the research process.

Within the different quality criteria governing qualitative research studies (COREQ and SRQR), one of the relevant aspects that ensure the transparency of the study is the description of the research team. Authors must include a section that describes the credentials of the research team at the time of the study (profession, experience in qualitative research, experience in psychiatry or mental health, reflexivity, previous position regarding the phenomenon of study, etc.).

They must also specify which team members conducted the interviews and who analyzed them, and who are the peer investigators who reviewed the investigation process (line 168). In addition, it is recommended to review both guides on quality criteria of qualitative studies to improve the methodological section of the article:

Consolidated criteria for reporting qualitative research (COREQ): a 32-item checklist for interviews and focus groups | The EQUATOR Network (equator-network.org)
Standards for reporting qualitative research: a synthesis of recommendations | The EQUATOR Network (equator-network.org)

REPLY

In the Methods section we have added that:  the areas included in the interview scheme were formulated following an attentive review of the relevant literature as well as experts’ opinion.

We have also clarified that: reflexive notes were also kept throughout the course of interviews and this helped the researchers to gain in depth understanding of the participants’ perceptions. These notes were also used to foster trustworthiness of the research process and this was described in the last paragraph of the section the 2.2.4. 

We have also added that:  the research team consisted of University teachers and experts from different scientific areas (e.g.  nursing, social work, social policy)  with experience in the care of people with mental illness and in qualitative research. Specifically, the team consists of five professors, with vast experience in designing and carrying out qualitative research. Most of them have long-term experience in working with vulnerable groups of people or with people with mental illness in the community and in psychiatric sites.

We have specified that: three members of the research team, who were well – trained in conducting interviews, interviewed the study participants. Although they had experience as social workers in mental health care, they did not work in any of the involved study sites and had no established relationship with the participants prior to the study. This was considered an advantage for the study as the participants could speak freely about their views on caregivers’ needs.

In addition the contribution of each member of the authoring team is presented at the end of the manuscript.

We have also added that, the study processes were evaluated by peer investigators – academics who were experienced in qualitative research.

We would like also to point out that we took into account the COnsolidated criteria for REporting Qualitative research checklist (COREQ), at least those that were applicable to our study. 

Between lines 140-148, the authors describe ethical criteria that should be placed in their own section called “ethical considerations” or similar, but not within the “data collection” section.

On line 147, they must include that the study has been approved by the ethics committee indicated on line 484 and not only by the participating entities.

REPLY

We have added the heading 2.2.5. Ethical Consideration  and we have specified that Prior to the commencement of the study, ethical approval was gained by the Ethical Committee of the Hellenic Mediterranean University as well as the  involved organizations (Ref. no 1029/2017-5-25).

In line 177, Table 1, should provide more data on the participants, in order to favor the transferability of the study. For example, the type of service each participant works on.

REPLY

We have added these data in Table 1.

In line 183, table 2, the two main topics called “caregivers´ feelings” and “caregivers´responsabilities” must go with their corresponding categories, as the following three topics appear. Therefore, the results 3.1. and 3.2. they should be rewritten according to the categories or subtopics obtained in the analysis. If there were no categories or sub-topics, it would indicate that the analysis has not been carried out correctly and / or that the information obtained is not enough to consider them independent topics.

REPLY

We have revisited the data and reformed the themes and subthemes accordingly. We have also made appropriate alterations in the results section as per suggestion. We hope that now the presentation of results is clearer.

From line 184 onwards, the explanation of the results obtained in each topic is scarce, the authors must describe each of the topics in more detail before providing narrative examples. In addition, the narrative examples must match the description. For example in line 188 they use the word "guilt" in the description and do not provide any narrative example of this feeling. They should review it on all topics. Before including narratives, it is not necessary for them to constantly repeat phrases such as "These were high through the following quotes." It is assumed that being a qualitative study, after the description of a result there will be narrative examples and including this phrase in all topics is repetitive. In addition, only the narrative of the participants should be in italics and between quotation marks, not the participant number or the explanatory notes that appear interspersed in some narratives (for example, in line 205 (the caregivers).

REPLY

We have made appropriate modifications to the results section. We have made corrections in line 205 and address all points raised by the reviewer.

Between lines 237-239, the authors name the categories identified from topic 3.3. differently than they do in table 2. The name of the categories must be the same. In addition, the categories included should be numbered to facilitate identification in the text. For example: 3.3.1. Financial and professional burdens. 3.3.2. Social constraits. And from then on follow this numbering. Each category, as well as each topic of all the results must be described more exhaustively before providing narrative examples.
The description of the category “Health deterioration” overlaps with the description of topic 3.1. “Caregivers' feelindgs”, providing similar results. Authors must differentiate results and not provide repeated results.

REPLY

In lines 237-239 we did not intent to name the categories but to refer to the areas of life that affected by the caregivers’ increased responsibilities.  We have rephrased by removing the a), b), c). We hope that this is more clear now. 

We have also modified the themes and subthemes by revisiting the data and the results section.

Between lines 308-310, narratives must be in italics.

REPLY

We have corrected that.

Between lines 337-342, narratives should not go with classifier points.
In the discussion section, references are missing. They should include the reference after each sentence and not just after several sentences in a row. For example, at the beginning of line 376, they must include the reference after this sentence: “More specifically, caregivers of psychotic patients experience intense feelings of loss, sadness, grief, and emotional exhaustion for patients with psychosis”. They should review the entire discussion to correct these errors, for example they do it again between lines 384-387.
The references in the discussion are not referenced correctly. For example, on line 378 you should replace [42, 43, 44] with [42-44]. In line 408 they must substitute [22, 51, 36,52, 53] for [22, 36, 51-53]. They should correct these kinds of mistakes throughout the discussion.

REPLY

We have corrected the above points as per suggestion.

The claims the authors make between lines 395-399 are incorrect. They cannot claim that there is an association between the caregiver role and the deterioration of the caregivers. Qualitative research does not establish associations between variables. They should reformulate indicating that the professionals describe this situation, but it is not an association in itself. The following sentence should be rephrased in a similar way and the authors should review the entire discussion and conclusions by rephrasing similar statements, such as lines 413-414: they cannot make that statement because it is not true that the results of their study indicate that caregivers develop health problems, but professionals describe this, which is different. The same, for example, in the conclusion of lines 463-465.

REPLY

We have rephrased to:  This study highlights that the increased responsibilities and demanding caring roles led to….  

We have also made appropriate corrections in the discussion section. 

In the study limitations section, line 447, they should indicate the impossibility of generalizing the study results due to the methodological design used. In addition, what is described between lines 454-459 does not correspond to the limitations of the study, but to what its study contributes and with future lines of research, so it should go in a specific section and not in study limitations.

REPLY

It was added that: Generalizability of the study results is not the expected outcome of the present study, as qualitative research designs do not aim at generalizing the results but at exploring the uniqueness of the phenomenon under investigation.

Finally, authors should check the references thoroughly, as there are many errors.

Considering the manuscript in a global way, I consider that the sample is scarce and that it should be expanded to provide greater depth to the information provided. Perhaps the little description of the topics and the lack of categories or subtopics in some of them may be the result of this. If this were possible, I think it is an interesting topic that deserves to be explored in greater depth.

REPLY

We corrected some errors in the References section. We have also referred to  the scarcity of the sample and we have made relevant recommendations for future research in the last section 4.2 Policy Implications and Recommendation for future research.

A cordial greeting.

Thank you very much!

The Authoring Team.

Round 2

Reviewer 2 Report

Congratulations on improving the manuscript. I hope it can now be published.

Regards.